# An empirical analysis based on a panel threshold model of the effect of Internet development on the efficiency of Chinese government public service supply

**Yongjie Wang[1], Yuan Liu[2]\*, Yuqun Hu[2]**

**1** School of Public Administration, Southwest Jiaotong University, Chengdu, Sichuan, China, **2** School of Public Policy and Administration, Xi'an Jiaotong University, Xi'an, Shaanxi, China

\* 675525246@qq.com

**Data Availability Statement:** All relevant data are within the paper and its Supporting Information files.

## Abstract

With the development of information technology, improving the efficiency of public services with the help of the Internet has become an important work of local governments. However, under different institutional environments, the impact mechanism of Internet development on the supply efficiency of government public services is still unclear. Based on China's interprovincial panel data from 2011 to 2019, this paper constructs a threshold effect model, sets the institutional environment as the threshold variable, and empirically analyzes the impact of Internet development on the supply efficiency of government public services. The results show that the difference in regional institutional environment will lead to the obvious threshold effect of Internet development on the supply efficiency of government public services: When the institutional environment is poor, the role of Internet development on the supply efficiency of government public services is not significant. With the improvement of the institutional environment, the role of Internet development in promoting the supply efficiency of government public services gradually appears, but the marginal intensity of promotion weakens. Compared with existing studies that mostly use linear models, this paper incorporates the institutional environment into the complex relationship between Internet development and government public service supply efficiency, and clarifies the role of the institutional environment in the process of Internet development affecting government public service supply efficiency and the non-linear relationship among the three. This paper reveals the mechanism of Internet development affecting the supply efficiency of government public services under different institutional environments and provides a new perspective for solving the shortage of public services.

## Introduction

Over the past 40 years of reform and opening up, China has made great achievements in economic development, but a series of public services related to people's well-being, which are

**Funding:** This research was supported by the National Social Science Fund of China [grant number 20VYJ027] The funders had no role in study design, data collection and analysis, decision to publish, or preparation of the manuscript.

**Competing interests:** The authors have declared that no competing interests exist.

provided by the government, have been increasingly unable to meet the growing service demands of the people [1, 2]. The overall supply of public services is characterized by insufficient quantity, poor quality, and structural imbalance [3], and the shortage of supply is a constraint on people's better life [4]. Report at the 19th National Congress of the Communist Party of China emphasizes that "we should insist on safeguarding and improving people's livelihood in development, and promoting people's well-being is the fundamental purpose of development. We should perform the government's redistributive regulating function and speed up the equalization of basic public services." [5]. The shortage of public service supply arises either because of insufficient inputs or inefficient supply. In recent years, China has actively promoted the reform of its political system, and all local governments have tried to solve the shortage of public service supply by increasing financial input, but they have not fundamentally solved this problem because they have ignored the needs of the people and sloppy management [6]. Therefore, to solve the shortage of public services in China, the focus should be on improving efficiency.

With the development of new-generation information technology, the Internet plays an important role in meeting the demand for diversified public services and reshaping the management model of public services. An increasing number of national and regional governments have begun to use the Internet to improve government operations and enhance the efficiency of public service supply [7, 8]. The People's Republic of China's national economic and social development of the fourteenth five-year plan clearly points out that "we should strengthen the construction of digital government and enhance the digitalization and intelligence of public services and social governance" [9], and the "Internet plus public services" has gradually become an important element for the Chinese government to promote the supply-side reform of public services and improve the efficiency of public service supply. However, as research progresses, some scholars point out that although the improvement of government service capacity is related to the use of information technology, it is not a key factor; instead, the coordination between the development of information technology and the regional institutional environment will affect the efficiency of government services more [10, 11]. Therefore, in the context of China's economic transformation at this stage, this paper attempts to answer the question of what effects the combination of Internet development and institutional environment will have on the efficiency of government public service supply. Is the enhancement of government public service supply efficiency by Internet development the same under different institutional environments? This paper expects to provide useful policy insights for China to enhance the efficiency of government public service provision through the Internet and build a service government that satisfies the people.

## Literature review

Public services are public goods, services or rights provided by the public sector, mainly the government, for common consumption and equal enjoyment by the general public according to the values of rights, equity, justice and universality, aiming to achieve the goal of maximizing social welfare [12]; public service efficiency is the comparative relationship between government inputs and comprehensive social benefits [13]. At present, scholars have conducted a large number of fruitful studies on how the government can achieve effective public service supply from both the social and economic development levels and have proposed specific strategies to improve it in terms of population density, urbanization level, and regional economic development level [14–16]. However, there is a relative lack of literature analyzing the efficiency of government public service supply from the perspective of Internet development, and the following two main aspects have been studied.

First, the impact of Internet development on the supply efficiency of government public services is analyzed from the perspective of supply costs. Internet development can significantly reduce the cost of government public services and effectively improve the efficiency of government public service supply by realizing communication and cooperation among the various supply subjects of public services [17]. At the same time, Internet development shortens the distance between the government and citizens, and people are able to move from long queues to online interactions, reducing human resource costs and thus improving the efficiency of service supply [7]. For example, Internet education will give people equitable access to superior educational resources and allow them to schedule their studies freely [18]; Internet transportation reduces traffic congestion, decreases traffic injuries, and transforms road traffic management systems into intelligent environments [19].

Second, the impact of Internet development on the supply efficiency of government public services is analyzed from the perspective of the supply system. The integration of the Internet and public services has increased the supply content and supply channels of public services and brought opportunities for innovating the public service system. Chen and Li proposed that the application of the Internet in the field of public services can effectively improve the supply efficiency of government public services in three ways: innovating the method of government public service supply, improving the level of public service sharing capacity, and providing rich and diverse public service channels [20]. In addition, Internet development has increased the transparency of public service processes and information, and the government can obtain timely information on people's needs and service feedback by adopting Internet technology, thus significantly improving the government's public service supply capacity and promoting service performance [21, 22].

The institutional environment is composed of political, social and legal ground rules outside the organization, which can regulate and guide the behavior of the organization [23, 24]. In the research related to e-government, scholars found that technology determines that everything is a "virtual beauty" if the institutional environment is not considered, and when the institutional environment is not mature, even the most advanced information technology may not be able to give full play to its capabilities [25]. A perfect institutional environment will produce constraints on the behavior of supply subjects, thus reducing the probability of invalid transactions and transaction costs, while an imperfect institutional environment not only lacks constraints on supply subjects but also may facilitate invalid transactions, resulting in high transaction costs [26]. Zhou and Dong analyzed from an organizational perspective and found that the institutional environment determines government behavior, and the better the institutional environment is, the higher the level of the local government's supply of public goods [27, 28]. Cai proposed that the institutional environment can effectively influence government public service performance by constructing a theoretical framework of institutional environment-institutional performance [29].

In summary, most of the existing studies tend to analyze the impact of Internet development or institutional environment on the efficiency of government public service supply, and individual studies have made normative analyses of the impact of Internet development on government public service supply under different institutional environments, but fewer studies have placed Internet development, institutional environment and government public service supply efficiency under the same framework and explored the relationship between Internet development under different institutional environments. However, fewer studies have placed Internet development, institutional environment and efficiency of government public service provision under the same framework and explored the specific mechanisms and heterogeneous effects between Internet development and efficiency of government public service supply in different institutional environments. Moreover, most of the current studies use linear

models and do not take into account the fact that the relationship among the three may be nonlinear and that there may be a threshold effect on the relationship as the level of institutional environment changes. This paper incorporates the institutional environment into the complex relationship between Internet development and government public service supply efficiency by constructing a panel threshold model. On the one hand, it explores the role of the institutional environment in the process of Internet development affecting government public service supply efficiency; on the other hand, it explores the nonlinear relationship between Internet development, the institutional environment, and government public service supply efficiency under different levels of the institutional environment.

## Research design

### Model setting

The impact of Internet development on the efficiency of government public service provision may not necessarily show a simple linear relationship; it may perhaps show a nonlinear relationship depending on the level of the institutional environment. To analyze this issue, this paper intends to conduct an empirical study using a panel threshold model.

Assuming a balanced panel dataset of $\{y_{it}, q_{it}, x_{it}: 1 \leq i \leq I, 1 \leq t \leq T\}$, the single threshold model can be expressed as

$$y_{it} = u_i + \beta_1 x_{it} \times I(q_{it} \leq \gamma) + \beta_2 x_{it} \times I(q_{it} > \gamma) + \varepsilon_{it} \tag{1}$$

In Eq (1), $i$ denotes region, and $t$ denotes time; $y_{it}$ and $x_{it}$ are the explanatory and explanatory variables, respectively; $q_{it}$ is the threshold variable, and $\gamma$ is the threshold value to be estimated; $I(\cdot)$ represents the indicative function, whose value takes 1 or 0 depending on the truth of the expression in parentheses; $u_i$ is the individual fixed effect; $\varepsilon_{it}$ is the random disturbance term; and $\beta_1$ and $\beta_2$ are the variable coefficients, with $\beta_1 \neq \beta_2$ indicating the presence of a threshold effect.

The above model assumes the existence of only a single threshold, but in reality, there may be two or more thresholds. Taking the existence of two thresholds as an example, the dual threshold model is set as follows.

$$y_{it} = u_i + \beta_1 x_{it} \times I(q_{it} \leq \gamma_1) + \beta_2 x_{it} \times I(\gamma_1 < q_{it} \leq \gamma_2) + \beta_3 x_{it} \times I(q_{it} > \gamma_2) + \varepsilon_{it} \tag{2}$$

In Eq (2), $\gamma_1 < \gamma_2$, and the meanings of other codes are the same as in (1). Based on the dual threshold model, this paper establishes a dual threshold model with the efficiency of government public service supply as the explanatory variable, the level of Internet development as the core explanatory variable, and the institutional environment as the threshold variable, which is used to explore the difference in the impact of the level of Internet development on the efficiency of government public service supply when the level of institutional environment is greater or less than a specific threshold. The specific model settings are as follows.

$$FW_{it} = u_i + \partial H_{it} + \beta_1 HLW_{it} \times I(ZD_{it} \leq \gamma_1) + \beta_2 HLW_{it} \times I(\gamma_1 < ZD_{it} < \gamma_2) + \beta_3 HLW_{it} \times I(ZD_{it} > \gamma_2) + \varepsilon_{it} \tag{3}$$

In Eq (3), the explanatory variable $FW_{it}$ indicates the efficiency of government public service supply; the core explanatory variable $HLW_{it}$ indicates the level of Internet development; the threshold variable $ZD_{it}$ indicates the level of institutional environment; and $H_{it}$ indicates a set of control variables that influence the efficiency of government public service supply, including human capital (RL), population density (MD), population structure (RK), urbanization level (CZ), regional economic level (RJ), and fiscal autonomy (ZZ).

## Data collection and analysis method

This paper selects panel data of 31 provinces, autonomous regions and municipalities directly under the central government (hereafter referred to as municipalities) in China from 2011–2019 for analysis to explore the impact of the level of Internet development and institutional environment on the efficiency of government public service supply. The data are mainly obtained from the China Statistical Yearbook, the China Population and Employment Statistical Yearbook, the China Marketization Index Report by Province, and the statistical yearbooks of each province, autonomous region, and municipality. After collecting the data, we mainly used STATA 15.0 software to process and analyze the data.

## Indicator selection

**Explanatory variable.** The efficiency of government public service supply (FW) is the explanatory variable. Government public service supply is a complex and systematic project involving multilevel and multifaceted public services. Based on the evaluation indexes of government public service supply efficiency referred to Hu et al. [30], this paper ultimately formed an evaluation index system of government public service supply efficiency involving infrastructure, public education, health care, culture and media, environmental protection and social security.

In terms of input indicators, since the main body of public service supply is the government and the government's resource input is reflected in human resource input and financial resource input, we choose to express the government's human resource input in terms of the ratio of public management-related employment to total regional employment in each region and the government's financial input in terms of the ratio of general public service expenditures to total government general public budget expenditures; in terms of output indicators, we mainly include six major areas: infrastructure, public education, health care, culture and media, environmental protection, and social security. Because of the multi-input-multioutput scenario, this paper adopts DEA method to measure the efficiency of public service supply of provincial, autonomous region and directly administered municipal governments by referring to relevant literature [31–34]. The government public service efficiency evaluation index system is shown in Table 1.

**Core explanatory variable.** The level of Internet development (HLW) is the core explanatory variable. At present, there is no unified measurement method for Internet development level. This paper draws on the Internet development level evaluation index system established by Li and Zhou [35], combines the current situation of China's Internet development, considers the availability and operability of data, and constructs an evaluation system for China's interprovincial Internet development level from three aspects: Internet usage, Internet infrastructure and Internet information resources, as shown in Table 2. This paper mainly adopts the entropy value method to assign weights to the evaluation indicators and then measures the Internet development level according to the weights of each indicator. The entropy value method, as an objective assignment method, can avoid the overlap of multiple indicator variables and reflect the utility value of the indicator entropy value, and the larger the weight is, the greater the influence of the indicator on the system [36].

**Threshold variable.** The institutional environment (ZD) is the threshold variable in this paper. Regarding the measurement of institutional environment indicators, the current views in academia are not uniform. In this paper, following the approach of Yang and Wang [37], we adopt the overall marketization index in the marketization index system of each region in China compiled by Wang et al., which is currently widely used, to represent the institutional environment. This index measures the marketization level of each region in five aspects:

**Table 1. Government public service supply efficiency evaluation index system.**

| Decision-Making Dimension | Indicator Dimension | Data Dimension |
|---|---|---|
| Input Indicators | Human Resource Input | Number of public management-related employees/Total number of regional employees |
| | Financial Resources Input | General public service expenditures/Total government general public budget expenditures |
| Output Indicators | Infrastructure | Water consumption per capita for residential use |
| | | Public transportation vehicles per 10,000 people |
| | Public Education | Number of general elementary school teachers as a percentage of the total population |
| | | Number of teachers in general secondary schools as a percentage of the total population |
| | Health Care | Number of medical and health institutions per 10,000 people |
| | | Number of beds in medical and health institutions per 10,000 people |
| | Culture and Media | (Art performance attendance + museum attendance + library book and literature lending attendance)/Total regional population |
| | Environmental Protection | Daily treatment capacity of urban sewage |
| | | Domestic garbage removal volume |
| | Social Security | Number of pension insurance participants per 10,000 people |
| | | Number of medical insurance participants per 10,000 people |

government-market relationship, nonstate economy, product market, factor market, and legal institutional environment. The higher the value of the marketization index is, the higher the marketization level of the region and the better the institutional environment.

**Control variables.** In this paper, control variables are included in the model to analyze the impact of Internet development on the efficiency of government public service supply under the condition of controlling some factors to avoid endogeneity problems caused by omitting relevant explanatory variables. The control variables are ① human capital level (RL), which is reflected by the average years of education of the regional population, with the average years of education of the population = (elementary school population*6 + middle school population*9 + high school and secondary school population*12 + college and above population*16)/total population; ② demographic structure (RK), using the juvenile dependency ratio, to examine the impact of demographic changes on the efficiency of government public service supply; ③ urbanization level (CZ), expressed as the proportion of the urban population at the end of the year; ④ population density (MD), calculated by dividing each province's year-end resident population by its land area; ⑤ fiscal autonomy (ZZ), measured by the ratio of revenue within the general public budget to expenditures within the general public budget; and ⑥ regional economic development level (RJ), expressed as GDP per capita. The descriptive statistics of the variables are shown in Table 3. From 2011 to 2019, the

**Table 2. Internet development level evaluation index system.**

| First-Level Indicators | Second-Level Indicators | Third-Level Indicators |
|---|---|---|
| Internet Development Level | Internet Usage | Internet broadband access users |
| | | Cell phone penetration rate |
| | Internet Infrastructure | Internet broadband access ports |
| | | Length of long-distance fiber optic cable |
| | Internet Information Resources | Number of web pages |
| | | Number of domain names |

**Table 3. Results of descriptive statistics of variables.**

| Variable | Sample size | Mean | Standard deviation | Min. | Max. |
|---|---|---|---|---|---|
| Efficiency of government public service supply (FW) | 279 | 0.941 | 0.079 | 0.67 | 1 |
| Level of Internet development (HLW) | 279 | 0.032 | 0.042 | 0.002 | 0.232 |
| Institutional environment (ZD) | 279 | 6.481 | 2.170 | -0.23 | 11.146 |
| Human capital (RL) | 279 | 9.667 | 0.791 | 7.463 | 12.936 |
| Demographic structure (RK) | 279 | 22.882 | 6.351 | 9.9 | 38.4 |
| Level of urbanization (CZ) | 279 | 0.566 | 0.134 | 0.222 | 0.942 |
| Population density (MD) | 279 | 0.045 | 0.070 | 0.0003 | 0.391 |
| Fiscal autonomy (ZZ) | 279 | 0.491 | 0.200 | 0.072 | 0.931 |
| Regional economic development level (RJ) | 279 | 10.745 | 0.442 | 9.690 | 12.011 |

efficiency of government public service provision in 31 Chinese provinces, autonomous regions and municipalities is relatively high, with a mean value of 0.941, but the minimum value is only 0.67, and the maximum value has reached 1, indicating that there may be large differences within different regions. The level of internet development in China is far from adequate, with a mean value of 0.032, which is far below the medium level, while the institutional environment in China is generally seen as basically reaching the medium level, but the gap between the maximum values also indicates large differences between different regions.

## Results and analysis

### Panel unit root test and cointegration test

In order to avoid the occurrence of pseudo regression, the variables were first tested for unit roots before regression analysis. In this study, both LLC and IPS were used to test each variable, and the results are shown in Table 4. All variables were significant at the 10% level, indicating that each variable was in a stationary state. However, the regression of univariate variables without cointegration is still a pseudo regression, so we used the Pedroni method to conduct cointegration tests on the panel data of variables, as shown in Table 5. Each statistic was significant at the 1% level, so there was a stable equilibrium relationship among the variables in the long run, and regression analysis could be performed.

**Table 4. Unit root test results for each variable.**

| Variable | Statistical Quantities | | Result |
|---|---|---|---|
| | LLC Inspection | IPS Inspection | |
| FW | -5.3198*** | -3.5802*** | stable |
| HLW | -10.4344*** | -11.2039*** | stable |
| ZD | -5.3119*** | -6.7673*** | stable |
| RL | -7.4215*** | -4.1042*** | stable |
| RK | -6.7735*** | -2.2535** | stable |
| CZ | -7.6396*** | -180*** | stable |
| MD | -15.1784*** | -13.6366*** | stable |
| ZZ | -5.1749*** | -83.5420* | stable |
| RJ | -7.0635*** | -8.2809*** | stable |

Note

*, **, *** indicate statistical results are significant at the 10%, 5% and 1% confidence levels, respectively.

**Table 5. Panel data cointegration test.**

| Variable | Heterogeneous PP | Heterogeneous ADF | Homogeneous PP | Homogeneous ADF |
|---|---|---|---|---|
| Statistical Quantities | -58.0525 | -29.4736 | 107.5051 | 76.9518 |
| P values | 0.000 | 0.000 | 0.000 | 0.000 |

**Threshold effect test.** This study empirically analyzed the role of the institutional environment in the relationship between Internet development and the efficiency of government public service supply, using the level of the regional institutional environment as the threshold variable. To obtain the number of thresholds, we first conducted a threshold effect test on Model (3) to determine the specific form of the threshold model, and the results are shown in Table 6. Under the assumptions of single, double and triple thresholds, the F values obtained from the analysis of the threshold effect and the P values obtained from 500 iterations of sampling using the bootstrap method reveal that the single threshold passed the 10% level of significance, the double threshold passed the significance level test at the 1% level, and the triple threshold did not pass the significance test. Therefore, in studying the impact of Internet development on the efficiency of government public service supply under different institutional environments, the author used the double threshold model to conduct the analysis.

From Table 7 (i.e., the results of threshold estimates with 95% confidence intervals for the institutional environment as the threshold variable), it can be seen that the 2 threshold estimates of the double threshold are 4.550 and 5.480, respectively, and thus the institutional environment can be divided into three different intervals: the poor institutional environment region (ZD≤4.550), the better institutional environment region (4.550<ZD≤5.480), and the perfect institutional environment region (ZD>5.480). The variability of the impact of Internet development on the efficiency of government public service supply can be analyzed in these three different institutional environment regions.

Figs 1 and 2 are the likelihood ratio function diagrams of the two thresholds 4.550 and 5.480, wherein, the lowest point of LR statistics is the corresponding real threshold value, and the dotted line indicates that LR reference value at a given 5% level is 7.35, while the LR statistics of the two estimated thresholds are below the reference value, which cannot reject the null hypothesis that the threshold significantly exists. Therefore, it can be considered that the threshold value is true and effective.

## Threshold regression analysis

After analyzing the derived thresholds, this paper estimated the parameters of the double-threshold model, and the results are shown in Table 8. For the control variables, the effects of human capital level, population density and regional economic level on the efficiency of government public service supply were not significant. In contrast, population structure,

**Table 6. Results of the threshold effect test.**

| Number of thresholds | F values | P values | Bootstrap | Critical value | | |
|---|---|---|---|---|---|---|
| | | | | 1% | 5% | 10% |
| single threshold | 7.843* | 0.070 | 500 | 13.293 | 9.922 | 6.296 |
| double threshold | 8.695** | 0.014 | 500 | 9.505 | 4.942 | 3.112 |
| triple threshold | 0.000 | 0.362 | 500 | 0.000 | 0.000 | 0.000 |

Note

*, **, *** indicate statistical results are significant at the 10%, 5% and 1% confidence levels, respectively.

Table 7. Threshold estimates and confidence intervals.

| Threshold $\gamma$ | Threshold estimates | 95% confidence intervals |
|---|---|---|
| Threshold $\gamma_1$ | 4.550 | [3.450, 10.635] |
| Threshold $\gamma_2$ | 5.480 | [3.450, 10.635] |

urbanization level and fiscal autonomy all had significant positive effects on the efficiency of government public service provision. Among them, the urbanization level had the greatest effect on the efficiency of government public service supply, reaching 0.445, which indicates that as urbanization progresses, the improvement of people's living standards is conducive to the gradual improvement and optimization of government public services, which promotes the continuous improvement of government public service supply efficiency. The next factor was financial autonomy, with a coefficient of 0.111, indicating that higher financial autonomy will help improve the efficiency of local government public service supply. Demographic structure also had a positive effect on the efficiency of government public service supply, with a coefficient of 0.009, which indicates that the relaxation of China's fertility policy will continue to promote the improvement of government public service supply, thus contributing to the improvement of the efficiency of government public service supply.

The results of the threshold effect test show that there is a significant dual threshold of the institutional environment for the impact of Internet development on the efficiency of government public service supply. In Table 8, when the level of marketization lies below 4.550, Internet development has a negative effect on government public service supply efficiency. Although it does not pass the significance test, it also at least indicates that Internet development does not necessarily enhance government public service supply efficiency; rather, the coordinated coupling between the level of the Internet and the regional institutional environment should be considered, and in regions with a poor institutional environment, there is a

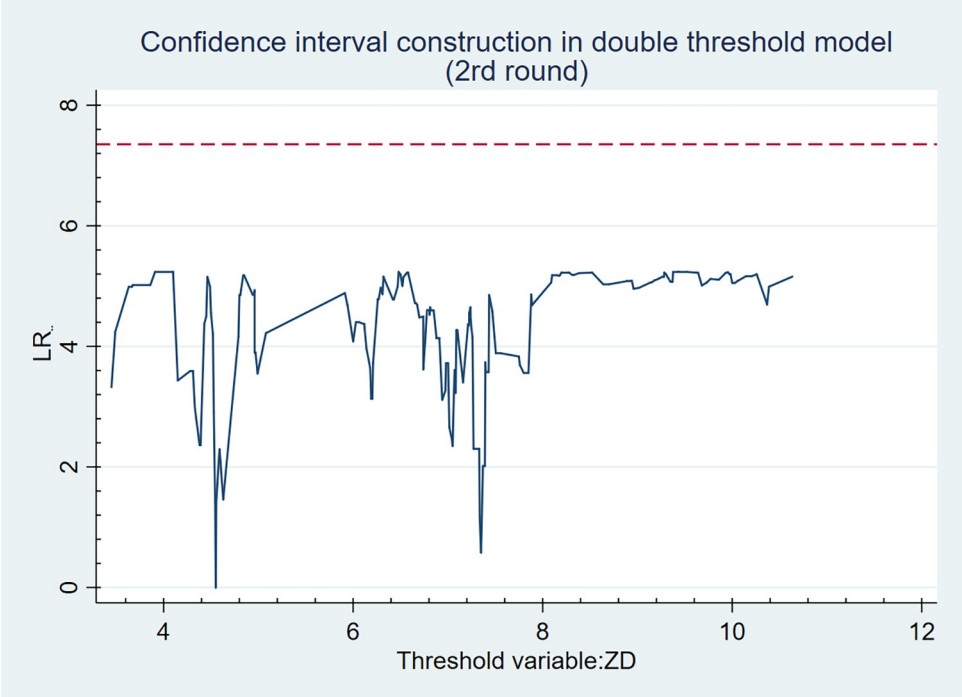

Fig 1. The likelihood ratio function diagram of the first threshold value.

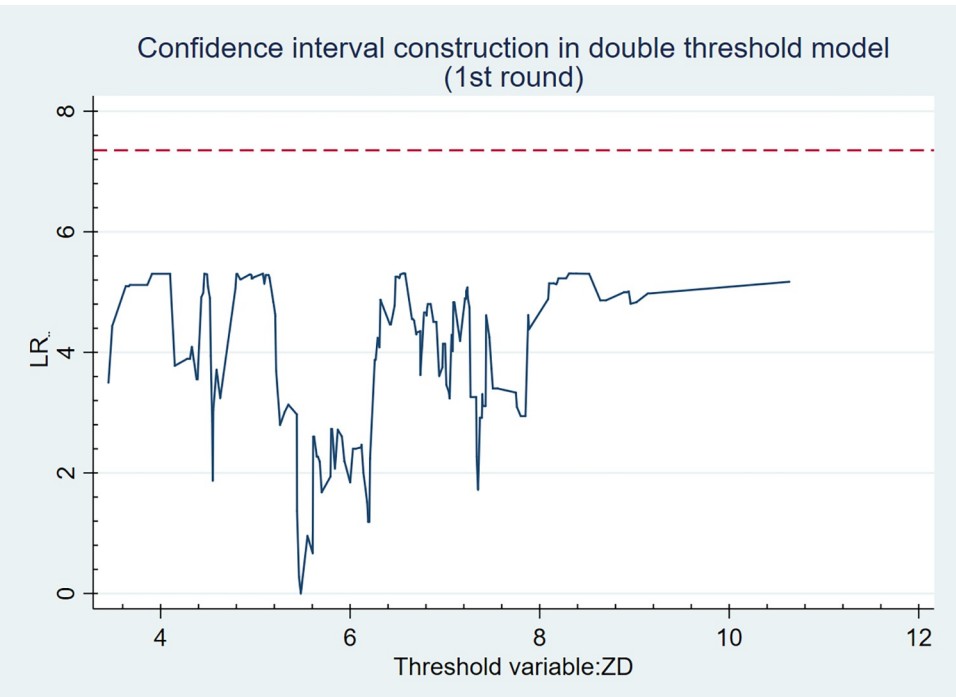

**Fig 2. The likelihood ratio function diagram of the second threshold value.**

high possibility that an increase in the level of Internet development will instead lead to a decrease in the efficiency of government public service provision. Internet development has a significant positive effect on the efficiency of government public service supply when the level of marketization is in the two intervals above 4.550. However, it can be found that the effect of Internet development on government public service supply efficiency is different in the two intervals of marketization levels above 4.550: with 5.480 as the cutoff point, when the marketization level is between 4.550 and 5.480, the positive contribution of Internet development to government public service supply efficiency is greater with the coefficient of action of 3.082; when the marketization level is higher than 5.480, the coefficient of Internet development on the efficiency of government public service supply decreases from 3.082 to 0.690 as the marketization level increases.

**Table 8. Estimation results of the parameters of the double threshold model.**

| Variables | Coefficient estimates | T-statistic | P values |
|---|---|---|---|
| RL | -0.00162 | -0.08 | 0.934 |
| RK | 0.00949*** | 4.99 | 0.000 |
| CZ | 0.445*** | 3.72 | 0.000 |
| MD | -0.0905 | -0.22 | 0.824 |
| ZZ | 0.111* | 1.74 | 0.082 |
| RJ | -0.0063 | -0.24 | 0.809 |
| HLW (ZD≤4.550) | -0.224 | -0.12 | 0.908 |
| HLW (4.550<ZD≤5.480) | 3.082*** | 2.97 | 0.003 |
| HLW (ZD>5.480) | 0.690*** | 3.52 | 0.001 |

Note

*, **, *** indicate statistical results are significant at the 10%, 5% and 1% confidence levels, respectively.

After the marketization level exceeds 4.550, there is always a positive promotion effect of the Internet development level on the efficiency of government public service supply, but the promotion intensity of the Internet development gradually weakens as the marketization level increases. This indicates that when the marketization level is low, the relevant network infrastructure is not sound, and the government's policy and institutional environment for using the Internet to supply public services is not standardized, which makes the government's awareness of using the Internet to supply services weak, and the role of Internet development in improving the efficiency of government public service supply is not obvious at this time. When the level of marketization reaches a certain level, the driving effect of Internet development becomes apparent, and the degree of promotion tends to diminish gradually, indicating that a higher level of marketization can prompt the government to make full use of the advantages of the Internet in terms of cross-time and space, multidirectional information dissemination, and technological spillover to improve the efficiency of government public service supply. When the marketization level of a region is high, the "Internet plus public services" in that region may have reached a certain critical stage of development, and the focus of public attention is not only on hardware facilities but also on the sense of public service experience and the in-depth integration of Internet services. However, since the integration of Internet technology in public services is not deep enough and the effective extension and expansion of public services is not enough, the efficiency of the Internet in promoting the government's public service supply will be reduced under such circumstances.

The 31 provinces, autonomous regions and municipalities of China were divided into 3 regions according to 2 thresholds of institutional environment (4.550, 5.480), and data for 2011 and 2019 are presented in Table 9.

As seen in Table 9, in 2011, there were 8, 6 and 17 provinces among 31 provinces, autonomous regions and municipalities in China that were in the low, higher and high marketability level regions, respectively, indicating that 74% of China's provinces had already crossed the first threshold value of marketability level in 2011, and the promotion effect of Internet development on the efficiency of government public service supply had already been revealed, among which 19% of the provinces were in the high-intensity promotion area and 54% of provinces were in the low-intensity promotion area. By 2019, only four of China's 31 provinces, autonomous regions and municipalities were in low-marketization-level regions, the same four were in higher-marketization-level regions, and 23 were in high-marketization-level regions, indicating that China's marketization level has improved relatively quickly over the past nine years, with the vast majority of provinces (87%) crossing the first threshold of marketization level and most provinces (74%) crossing the second threshold value of marketization level, while the four provinces having not yet crossed the first threshold value and finding themselves in a situation where the efficiency of Internet development on government public service supply was not significant are concentrated in the western region.

**Table 9. Regional distribution pattern of the level of marketization of provincial administrative units in 2011 and 2019.**

| Classification basis | 2011 | 2019 |
|---|---|---|
| Low-marketing-level regions (ZD≤4.550) | Inner Mongolia, Guizhou, Tibet, Shaanxi, Gansu, Qinghai, Ningxia, Xinjiang | Tibet, Gansu, Qinghai, Ningxia |
| Medium-marketing-level regions (4.550<ZD≤5.480) | Hebei, Shanxi, Heilongjiang, Guangxi, Hainan, Yunnan | Inner Mongolia, Hainan, Guizhou, Xinjiang |
| High-marketing-level regions (ZD>5.480) | Beijing, Tianjin, Liaoning, Jilin, Shanghai, Jiangsu, Zhejiang, Anhui, Fujian, Jiangxi, Shandong, Henan, Hubei, Hunan, Guangdong, Chongqing, Sichuan | Beijing, Tianjin, Hebei, Shanxi, Liaoning, Jilin, Heilongjiang, Shanghai, Jiangsu, Zhejiang, Anhui, Fujian, Jiangxi, Shandong, Henan, Hubei, Hunan, Guangdong, Guangxi, Chongqing, Sichuan, Yunnan, Shaanxi |

## Conclusions and implications

This study used interprovincial panel data from 2011–2019 and a nonlinear panel threshold model to conduct an in-depth study of the effect of Internet development on the efficiency of government public service supply under different institutional environments. The results found that the effect of Internet development on the efficiency of government public service supply differs under different institutional environments. When the level of marketization is in a low range, the effect of Internet development on the supply efficiency of government public services is not obvious, but when the level of marketization crosses a certain threshold, Internet development makes a significant contribution to the supply efficiency of government public services. However, as the level of marketization continues to climb, the role of Internet development in promoting the efficiency of government public service supply begins to weaken. Currently, all provinces in eastern China and central China and four provinces in western China—Chongqing, Sichuan, Yunnan, and Shaanxi—are already in the high-marketization-level range, while four of the remaining eight provinces in western China are in the medium-marketization-level range and the low-marketization-level ranges. Based on the above findings, this study proposes the following policy recommendations.

(1) The effect of Internet development on the efficiency of government public service supply differs from region to region depending on the level of the institutional environment. When formulating policies related to the improvement of public service supply efficiency, local governments should pay attention to the differences in the regional institutional environment, formulate different policies and take different measures. For example, in the eastern and central regions, where the marketization level is high and the institutional environment is relatively perfect, the promotion effect of Internet development on the government's public service supply efficiency is significant, so governments in these regions should increase their support for Internet development, while some provinces in the western region have a low marketization level and an imperfect institutional environment, so governments at all levels should first make efforts to improve the institutional environment, such as continuing to vigorously promote the regional marketization process, paying attention to the role of the market in the process of public service supply, introducing market competition mechanisms, etc., and then increase the investment in Internet development on this basis. At the same time, they should pay attention to the coordination and coupling between the level of Internet development and the institutional environment to avoid "ineffective development" caused by the lack of coordination.

(2) The Internet plays a significant role in improving the efficiency of government public service supply, and each local government should increase its investment in Internet development but should adopt differentiated strategies in response to the actual development of different regions. First, for regions with a low level of Internet development, the government should give full play to its guiding role, actively formulate policies for Internet development, expand the scope of government use of the Internet and improve people's ability to use the Internet; continuously increase the intensity of investment in the Internet, improve Internet infrastructure and information resources, and enhance information supply capacity; and build a collaborative development pattern for the Internet, focusing on infrastructure, information resources, and Internet applications to achieve high-quality development of the Internet. Second, for regions with a high level of Internet development, they should further broaden the development and application of the Internet in the field of public services, optimize the public service system using emerging technologies such as big data, the Internet, cloud computing and the Internet of Things, and accelerate the innovation of the public service supply model. Third, different development regions should base on their own Internet development advantages,

make up for their own development shortcomings, and improve Internet service capabilities, while the central government can provide policy inclination and financial and technical support to regions with lower levels of Internet development to promote their rapid development and continuously narrow the development gap between the eastern, central and western regions.

(3) Due to the lack of integration of "Internet plus public services", the role of Internet development in promoting the efficiency of government public service supply in regions with high marketization levels has begun to weaken, so local governments, while improving the level of the Internet, should realize the opening and sharing of relevant data and the interoperability of relevant resources in public service supply by creating public service network platforms, promoting the simplification of network processes, and formulating unified standard planning, operation specifications and supply guidelines, so as to continuously release the vitality and potential of the Internet, further expand the breadth and depth of integration between the Internet and public services such as education, medical and health care, elderly care and housekeeping services, and continuously enhance the role of Internet development in promoting the efficiency of government public service supply.

## Supporting information

**S1 Table. Study's minimal underlying data.**
(XLSX)

## Acknowledgments

We sincerely thank all the participants in this study.

## Author Contributions

**Conceptualization:** Yongjie Wang.

**Data curation:** Yuan Liu.

**Formal analysis:** Yongjie Wang.

**Funding acquisition:** Yongjie Wang.

**Methodology:** Yuan Liu.

**Software:** Yuan Liu.

**Supervision:** Yuqun Hu.

**Validation:** Yuqun Hu.

**Visualization:** Yuqun Hu.

**Writing – original draft:** Yuan Liu.

**Writing – review & editing:** Yuqun Hu.

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
