## [Decision Letter · Decision Letter 0]

14 Jun 2022

PONE-D-22-13139Can Internet development promote the efficiency of Chinese government's public service supply?—— An empirical analysis based on a panel threshold modelPLOS ONE

Dear Dr. yuan Liu,

Thank you for submitting your manuscript to PLOS ONE. After careful consideration, we feel that it has merit but does not fully meet PLOS ONE’s publication criteria as it currently stands. Therefore, we invite you to submit a revised version of the manuscript that addresses the points raised during the review process.

We look forward to receiving your revised manuscript.

Kind regards,

Carlos Alberto Zúniga-González, Ph.D

Academic Editor

PLOS ONE

Journal Requirements:

[We sincerely thank the National Social Science Fund of China for supporting this article. We also thank all the participants in this study. This research was supported by the National Social Science Fund of China [grant number 20VYJ027].]

 [This research was supported by the National Social Science Fund of China [grant number 20VYJ027]

The funders had no role in study design, data collection and analysis, decision to publish, or preparation of the manuscript.]

4. PLOS requires an ORCID iD for the corresponding author in Editorial Manager on papers submitted after December 6th, 2016. Please ensure that you have an ORCID iD and that it is validated in Editorial Manager. To do this, go to ‘Update my Information’ (in the upper left-hand corner of the main menu), and click on the Fetch/Validate link next to the ORCID field. This will take you to the ORCID site and allow you to create a new iD or authenticate a pre-existing iD in Editorial Manager. Please see the following video for instructions on linking an ORCID iD to your Editorial Manager account: https://www.youtube.com/watch?v=_xcclfuvtxQ.

Additional Editor Comments:

Dear authors, I am sharing the observations of the reviewers, which are minimal; however, I would like the number of references to increase. I suggest the following.

[1] Blanco-Orozco, N., Arce-Díaz, E., & Zúñiga-Gonzáles, C. (2015). Integral assessment (financial, economic, social, environmental and productivity) of using bagasse and fossil fuels in power generation in Nicaragua. Revista Tecnología en Marcha, 28(4), 94-107. DOI 10.18845/tm.v28i4.2447 https://publons.com/publon/32281799/

[2] Zuniga González, C. A. (2020). Total factor productivity growth in agriculture: Malmquist index analysis of 14 countries, 1979-2008. REICE: Revista Electrónica De Investigación En Ciencias Económicas, 8(16), 68–97. https://doi.org/10.5377/reice.v8i16.10661

[3] Zuniga-Gonzalez, Carlos Alberto (2021), “Total factor productivity in the INTA Chinandega rice variety”, Mendeley Data, V2, doi: 10.17632/76m7p7mvsg.2 https://data.mendeley.com/datasets/76m7p7mvsg/2

[4] González, C. A. Z. (2011). Technical efficiency of organic fertilizer in small farms of Nicaragua: 1998-2005. African Journal of Business Management, 5(3), 967-973. https://publons.com/publon/11272633/

[5] Dios-Palomares, R., Alcaide, D., Diz, J., Jurado, M., Prieto, A., Morantes, M., & Zúñiga, C. A. (2015). Analysis of the efficiency of farming systems in Latin America and the Caribbean considering environmental issues. Revista Cientifica, Facultad de Ciencias Veterinarias, Universidad del Zulia, 25(1), 43-50. https://publons.com/publon/3106827/

[6] Bermúdez-León, D. S., & Zúniga-González, C. A. (2016). Information and communication technologies (ICT) as a response to educational needs in rural areas in Nicaragua. Rev. Iberoam. Bioecon. Cambio Clim., 2(4), 563–574. https://doi.org/10.5377/ribcc.v2i4.5931

Reviewers' comments:

Reviewer's Responses to Questions

**Comments to the Author**

1. Is the manuscript technically sound, and do the data support the conclusions?

Reviewer #1: Yes

Reviewer #2: Yes

2. Has the statistical analysis been performed appropriately and rigorously? 

Reviewer #1: Yes

Reviewer #2: Yes

3. Have the authors made all data underlying the findings in their manuscript fully available?

Reviewer #1: Yes

Reviewer #2: Yes

4. Is the manuscript presented in an intelligible fashion and written in standard English?

Reviewer #1: Yes

Reviewer #2: Yes

5. Review Comments to the Author

Reviewer #1: In this paper, the authors are proposed “Can Internet development promote the efficiency of Chinese government's public service supply?—— An empirical analysis based on a panel threshold model”

The strengths of the paper are that it is well structured, the description of the related work is well done and that results are extensively compared to results of the similar research.

Minor revisions:

1. Authors change the title of the manuscript “An empirical analysis based on a panel threshold model of Chinese government public service supply with internet development efficiency”.

2. Authors draw a graphical abstract of the model.

3. Proofread the entire manuscript.

Reviewer #2: In my opinion, this article presents an analysis based on statistical analysis of the impact of the Internet on the efficiency of public services in various environments. In addition, the methodology and statistical analysis was presented with warmth and scientific precision.

6. PLOS authors have the option to publish the peer review history of their article (what does this mean?). If published, this will include your full peer review and any attached files.

Reviewer #1: No

Reviewer #2: **Yes: **Napoleon Vicente Blanco Orozco

---

## [Author Response · Author response to Decision Letter 0]

16 Jun 2022

We sincerely appreciate your valuable and constructive comments on our manuscript (Manuscript Number:PONE-D-22-13139). We have studied the comments carefully and have made revisions. According to these comments, we have revised our manuscript very carefully. All changed parts are highlighted in yellow in the revised manuscript. The point-by-point answers to all comments are listed as follows. 

The following are the comments and suggestions pointed out by Reviewers.

Point 1: Authors change the title of the manuscript “An empirical analysis based on a panel threshold model of Chinese government public service supply with internet development efficiency”.

Response 1: Thank you for your comments. In the revised version, we have revised the title of the article as “An empirical analysis based on a panel threshold model of the Effect of Internet Development on the Efficiency of Chinese Government Public Service Supply”(line 1-4).

Point 2: Authors draw a graphical abstract of the model.

Response 2: Thank you for your comments. In the revised version, we have supplemented the likelihood ratio function graphs of the two thresholds (Fig. 1 and Fig. 2), and further explained the contents of the two graphs (line 318-326). The pictures will be uploaded separately in TIFF format.

Point 3: Proofread the entire manuscript.

Response 3: Thank you for your comments. In the revised version, we have carefully Proofread the full text of the manuscript.

---

## [Decision Letter · Decision Letter 1]

30 Jun 2022

An empirical analysis based on a panel threshold model of the Effect of Internet Development on the Efficiency of Chinese Government Public Service Supply

PONE-D-22-13139R1

Dear Dr. Yuan Liu,

We’re pleased to inform you that your manuscript has been judged scientifically suitable for publication and will be formally accepted for publication once it meets all outstanding technical requirements.

Kind regards,

Carlos Alberto Zúniga-González, Ph.D

Academic Editor

PLOS ONE

Additional Editor Comments (optional):

Congratulations!!!!!!!!The manuscript has been substantially improved and comments from reviewers have been incorporated.

Reviewers' comments:

Reviewer's Responses to Questions

**Comments to the Author**

1. If the authors have adequately addressed your comments raised in a previous round of review and you feel that this manuscript is now acceptable for publication, you may indicate that here to bypass the “Comments to the Author” section, enter your conflict of interest statement in the “Confidential to Editor” section, and submit your "Accept" recommendation.

Reviewer #1: All comments have been addressed

2. Is the manuscript technically sound, and do the data support the conclusions?

Reviewer #1: Yes

3. Has the statistical analysis been performed appropriately and rigorously? 

Reviewer #1: Yes

4. Have the authors made all data underlying the findings in their manuscript fully available?

Reviewer #1: Yes

5. Is the manuscript presented in an intelligible fashion and written in standard English?

Reviewer #1: (No Response)

6. Review Comments to the Author

Reviewer #1: The strengths of the paper are that it is well structured, the description of the related work is well done and that results are extensively compared to results of the similar research.

Authors addressed all my comments.

7. PLOS authors have the option to publish the peer review history of their article (what does this mean?). If published, this will include your full peer review and any attached files.

Reviewer #1: No

---

## [Editor Report · Acceptance letter]

4 Jul 2022

PONE-D-22-13139R1 

An empirical analysis based on a panel threshold model of the Effect of Internet Development on the Efficiency of Chinese Government Public Service Supply 

Dear Dr. Liu:

I'm pleased to inform you that your manuscript has been deemed suitable for publication in PLOS ONE. Congratulations! Your manuscript is now with our production department. 

Kind regards, 

on behalf of

Dr. Prof. Carlos Alberto Zúniga-González 

Academic Editor

PLOS ONE